

# Evolution of sexual size dimorphism in the wing musculature of *Drosophila*

Claire B. Tracy, Janet Nguyen, Rayna Abraham and Troy R. Shirangi

Department of Biology, Villanova University, Villanova, PA, United States of America

## ABSTRACT

Male courtship songs in *Drosophila* are exceedingly diverse across species. While much of this variation is understood to have evolved from changes in the central nervous system, evolutionary transitions in the wing muscles that control the song may have also contributed to song diversity. Here, focusing on a group of four wing muscles that are known to influence courtship song in *Drosophila melanogaster*, we investigate the evolutionary history of wing muscle anatomy of males and females from 19 *Drosophila* species. We find that three of the wing muscles have evolved sexual dimorphisms in size multiple independent times, whereas one has remained monomorphic in the phylogeny. These data suggest that evolutionary changes in wing muscle anatomy may have contributed to species variation in sexually dimorphic wing-based behaviors, such as courtship song. Moreover, wing muscles appear to differ in their propensity to evolve size dimorphisms, which may reflect variation in the functional constraints acting upon different wing muscles.

## INTRODUCTION

During courtship, males of most species in the genus *Drosophila* extend and vibrate a wing to produce a 'song.' Courtship songs are impressively variable across *Drosophila* species (*Tomaru & Yamada, 2011*) so much so that males of each species are thought to sing a unique, species-specific song. Song diversity most likely evolved by changes in the neural circuits that pattern the song in males; however, anatomical or physiological changes in the wing muscles that generate the song may have also contributed to song variation. The extent to which evolutionary transitions in the wing periphery may have influenced the evolution of courtship song is currently unclear.

Dipteran wing muscles are housed within the fly's thorax and fall into two anatomical and functional categories (reviewed in *Dickinson & Tu, 1997*). Contractions of the large indirect wing muscles power wing movements during flight and courtship song. The fine control of these behaviors, however, is modulated by a set of 18 small control wing muscles that line the lateral wall of the thorax. Most control wing muscles insert into one of four sclerites (i.e., small cuticular plates) at the wing hinge and regulate wing movement by altering the mechanical properties of the hinge (reviewed in *Deora, Gundiah & Sane, 2017*). Control muscles that insert directly into the wing hinge are classified into four

Corresponding author
Troy R. Shirangi,
troy.shirangi@villanova.edu

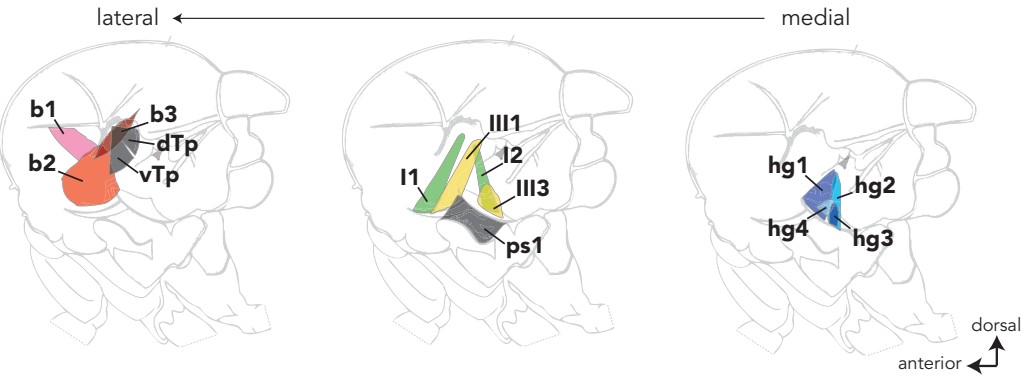

**Figure 1  The locations of most control wing muscles in the *Drosophila* hemithorax.** The control wing muscles are classified according to the sclerite at the wing hinge in which they insert. The b1–3 muscles (red) are associated with the basalare; the I1 and I2 muscles (green) target the first pterale; the III1 and III3 muscles insert into the third pterale; the hg1–4 (blue) muscles are associated with the posterior notal wing process. Sclerites at the wing hinge are not shown. Hemithoraces are arranged to show the lateral-most muscles on the left and the medial-most muscles on the right.

groups (Fig. 1) based on the sclerite into which they insert (i.e., basalare, first and third pterale, and the posterior notal wing process).

How the control wing muscles influence the generation of *Drosophila* song has been investigated most thoroughly in *Drosophila* (*D.*) *melanogaster* (*Ewing, 1977*; *Ewing, 1979*; *Shirangi, Stern & Truman, 2013*; *O'Sullivan et al., 2018*). The song of *D. melanogaster* males consists of trains of pulses and bouts of continuous tone called pulse song and sine song, respectively. Our previous work identified a control wing muscle in *D. melanogaster* called hg1 whose activity was found to influence the generation of sine song (*Shirangi, Stern & Truman, 2013*). Notably, of the 13 control wing muscles analyzed in *D. melanogaster*, hg1 was found to be the only one that is larger in males than in females. Males with a feminized hg1 muscle have an hg1 muscle that is reduced in size and sing sine song more quietly than normal males. Moreover, *D. melanogaster* females are relatively less receptive to courtship from males with a feminized hg1 muscle, suggesting that the size of the hg1 muscle and the volume with which the male sings sine song contributes to the female's willingness to mate.

Sine song has evolved a great deal across *Drosophila* species (*Tomaru & Yamada, 2011*). For instance, some species like *D. santomea* and *D. yakuba* have lost sine song during male courtship (*Watson, Rodewald & Coyne, 2007*), whereas other species like *D. mauritiana* and *D. simulans* display quantitative variation in certain aspects of sine song (*Ding et al., 2016*). This prompted us to investigate the evolutionary history of hg1's sexual size dimorphism in *Drosophila*. Our analysis revealed that hg1 and other control wing muscles have undergone a notable number of evolutionary transitions in sexual size dimorphism. This pattern of evolutionary change suggests that transformations in the *Drosophila* wing musculature may have contributed to species variation in sexually dimorphic wing-based behaviors, such as courtship song. Furthermore, we found that some muscles display more variation

in sexual size dimorphism than others, suggesting that wing muscles vary in the selective pressures that drive the evolution of size dimorphisms.

## METHODS AND MATERIALS

### Fly stocks

Flies were reared on standard cornmeal and molasses food at room temperature. Most stocks of species other than *D. melanogaster* were obtained from the *Drosophila* Species Stock Center (now located at Cornell University, College of Agriculture and Life Science).

### Measurement of wing muscle volume

To visualize and measure the volumes of wing muscles, hemithoraces from males or females of each species were dissected in PBS and fixed in 4% paraformaldehyde (buffered in PBS) for approximately 50 min at room temperature. To effectively stain the control wing muscles lining the lateral wall of the thorax, we removed the six large fibers of the dorsal longitudinal muscle after fixation. Fixed tissues were washed in PBS-TX (PBS with 1% Triton X-100), placed in PBS-TX containing Texas Red-X phalloidin (1:50; Life Technologies), and incubated for 3 to 4 days at 4 °C. Tissues were subsequently washed all day in PBS-TX, placed onto poly-lysine-coated coverslips, dehydrated through an ethanol series, cleared in xylenes, and mounted in DPX (Sigma-Aldrich). Tissues were imaged on a Zeiss LSM 510 confocal microscope at 10X with optical sections at 1 μm intervals. Confocal stacks of phalloidin-stained hemithoraces were imported into Amira (Thermo Scientific). The wing muscles were segmented and reconstructed by selecting and assigning pixels through the confocal series to labels of their respective wing muscle. Amira was used to measure muscle volume using the appropriate voxel dimensions (in mm).

Given species variation in overall body size between males and females, we measured a 'normalized' hg1, hg2, hg3, and hg4 muscle volume for each dissected hemithorax. The raw muscle volume of each hg muscle, the first basalare muscle, and the dorsal and ventral Tergopleural muscles were measured for each hemithorax. A normalized muscle volume was obtained by dividing the volume of each hg muscle for each hemithorax by the total muscle volume (i.e., hg1 + hg2 + hg3 + hg4 + b1 + d-Tp + v-Tp) of the respective hemithorax. The normalized muscle volumes for males and females of each species were used to calculate the Cohen's *d* effect size score (*Cohen, 1988*) to estimate the magnitude of the difference in normalized muscle volumes between males and females. The effect size difference between males and females for each species was used to create a categorical dataset, categorizing each value as either "male enlarged," "female enlarged," or "monomorphic." Species with a Cohen's $d \geq 1.5$ were considered to be dimorphic, whereas species with a Cohen's $d < 1.5$ were categorized as monomorphic. A Cohen's $d \geq 1.5$ means that the mean normalized muscle volume of one sex is above the 90th percentile of the opposite sex's normalized muscle volume distribution.

### Phylogenetic estimation

The phylogenetic estimate for the genus *Drosophila* utilizes sequence data available in the GenBank Nucleotide Database (*Clark et al., 2016*) for six nuclear genes (ADH, amyrel,

Ddc, Gphd, Sod, and XDH; Tables S1 and S2). Phylogenetic relationships were estimated using taxon sampling similar to that of described previously (*Russo et al., 2013*) for the *D. montium*, *D. melanogaster*, and *D. ananassae* subgroups. The phylogeny was estimated using maximum likelihood in IQ-TREE v. 1.6.1 (*Nguyen et al., 2015*) from a concatenated alignment of the six genes. Each gene was partitioned by codon position to estimate models of sequence evolution using ModelFinder (*Kalyaanamoorthy et al., 2017*) in IQ-TREE, substitution models were chosen based on the Bayesian Information Criterion (BIC). The IQ-TREE analysis was run with 1000 ultrafast bootstrap replicates (*Minh, Nguyen & Von Haeseler, 2013*). Subsequent to estimation of the phylogeny the tree was pruned to include only taxa for which muscle volume data was available for the ancestral state reconstruction.

The phylogenetic analysis resulted in a well-supported phylogeny with relationships matching those found in previous studies (e.g., *Russo, Takezaki & Nei, 1995*). The backbone of the tree was largely supported, however some of the groups are not well supported in the maximum likelihood tree (Figs. S1, S2). As the majority of the backbone relationships were well supported, we pruned the phylogeny to include only species for which we have muscle volume data. The pruned tree is largely supported with the only unsupported nodes falling within the *melanogaster* species group (Fig. S3). While not all nodes within the *melanogaster* species group are well supported, the relationships recovered match previous studies of the *Drosophila* phylogeny.

### Ancestral state reconstruction

An ancestral state reconstruction on the categorical dataset was run in R-Studio (*R Core Team, 2017*) using the ancestral character estimation tool in the package 'ape' (*Paradis & Schliep, 2019*). This function uses maximum likelihood estimation in a two-pass manner to work from the values at each tip back to the root of the tree, then working from the root back to the tips to determine the likelihood of each state at each node. Likelihood values for each node were then mapped onto the phylogenetic tree in pie chart format to indicate the relative likelihood of dimorphic (male-enlarged or female-enlarged) vs monomorphic being the ancestral state at that specific node.

## RESULTS

### The evolutionary history of hg1′s sexual size dimorphism

Given the role of hg1 and its size dimorphism in patterning *D. melanogaster* courtship song (*Shirangi, Stern & Truman, 2013*), we sought to determine the history of hg1′s sexual dimorphism in the *Drosophila* phylogeny: when did hg1′s size dimorphism emerge in *Drosophila*, and how stable has it been since it arose? We thus analyzed the volume of the hg1 muscle from males and females of 16 species within the *melanogaster* species group, which emerged ∼25 Mya (*Russo, Takezaki & Nei, 1995*), and 3 outgroup species (*D. obscura*, *D. willistoni*, and *D. virilis*). Using these species, we estimated a phylogeny using maximum likelihood methods, identified the species with a dimorphically or monomorphically sized hg1 muscle, then employed methods in ancestral state reconstruction to estimate the probability of a dimorphic or monomorphic hg1 muscle in the ancestral nodes of the phylogeny.

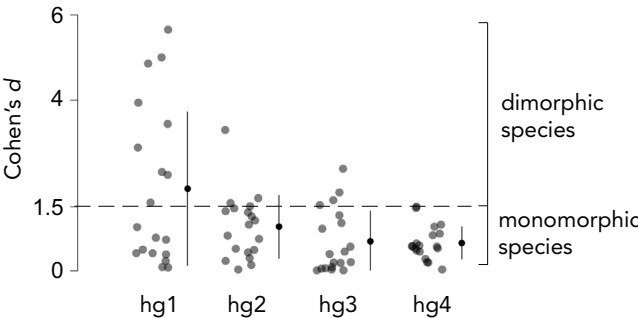

**Figure 2** **Effect size of the difference in normalized wing muscle size between males and females of 19** ***Drosophila* species.** To categorize each species as having a dimorphically sized or monomorphically sized hg wing muscle, we quantified the magnitude of the difference in normalized wing muscle volume between males and females of each species using Cohen's *d* effect size. Species that had a Cohen's *d* value of 1.5 or greater were categorized as "dimorphic," whereas those below 1.5 were binned as "monomorphic." The magnitude of the muscle size difference between males and females was generally lower for hg2–4 compared to hg1. Individual points, the mean and SD is shown for each muscle.

Overall body size typically differs between *Drosophila* males and females, and the magnitude by which males and females differ in size varies across species. We thus measured the *relative* hg1 muscle volume by normalizing the raw hg1 volume of each fly to the total volume of seven wing muscles. To determine if a species has a dimorphically sized hg1 muscle, we measured the magnitude, i.e., Cohen's *d* effect size (*Cohen, 1988*), of the difference in normalized hg1 volume between males and females of each species (Figs. 2 and 3). A species was categorized as "dimorphic" if the male's mean hg1 volume fell above the 90th percentile of the female's hg1 muscle volume distribution. Such species have a Cohen's *d* value of 1.5 or greater (*Cohen, 1988*). Species with a Cohen's *d* value below 1.5 were deemed "monomorphic."

The *melanogaster* species group consists of several subgroups that form three main clades: the *ananassae* species subgroup, the *montium* species subgroup, and the Oriental lineage, which includes the *melanogaster*, *suzukii*, *eugracilis*, and *elegans* subgroups. Ancestral state reconstruction of hg1's sexual size dimorphism (Fig. 3, Fig. S1, Table S1) indicates that a sexually dimorphic, male-enlarged hg1 muscle was most likely present in the ancestor of the *melanogaster*, *suzukii*, and *eugracilis* subgroups (probability of a dimorphic ancestor = 0.839). Ancestral nodes within the *melanogaster*, *suzukii*, and *eugracilis* subgroups remain strongly dimorphic (probability of dimorphic ancestor ≥ 0.859), indicating that the size dimorphism was lost independently twice in *D. eugracilis* and *D. orena*, both of which have a monomorphic hg1 muscle. Nearly all species studied (8/9 species) outside of the *melanogaster*, *suzukii*, and *eugracilis* subgroups have a monomorphically sized hg1 muscle, suggesting that the dimorphism may have first emerged in the ancestor of the *melanogaster*, *suzukii*, and *eugracilis* subgroups. However, given that *D. fuyamai* has a strongly dimorphic hg1 muscle, it is possible that a dimorphic hg1 muscle emerged earlier in the phylogeny in the ancestor of the Oriental lineage, or independently in the lineage leading to *D. fuyamai*. Nevertheless, our results indicate that a sexually dimorphic hg1 muscle arose within the

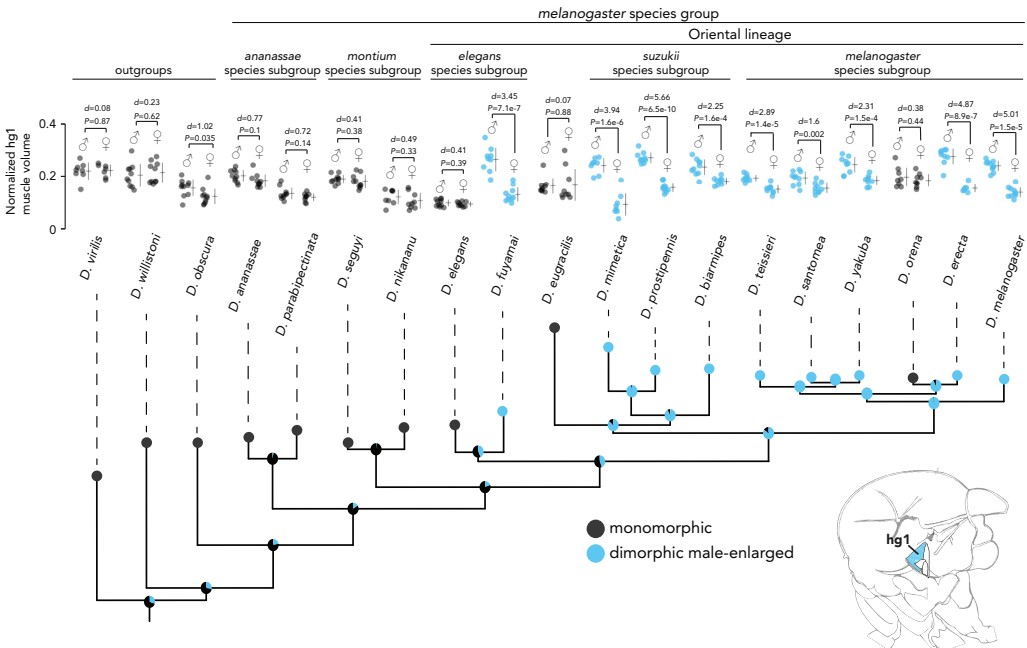

**Figure 3** **Evolutionary history of hg1's sexual size dimorphism in *Drosophila*.** Maximum likelihood tree showing phylogenetic relationships among 19 *Drosophila* species. Circles at internal nodes represent ancestral probabilities of a dimorphic (blue) or monomorphic (black) hg1 muscle inferred using maximum likelihood. Probability values at each node are reported in Fig. S1 and Table S1. The normalized volumes of the hg1 wing muscle in males and females of each species is also shown. Individual points, the mean and SD are given for each. N ≥ 10 hemithoraces for most groups. Data of species with a dimorphic hg1 muscle (i.e., Cohen's $d \geq 1.5$) are in blue, whereas data of species with a monomorphic hg1 muscle (i.e., Cohen's $d < 1.5$) are shown in black. P values were obtained using a standard Student's $t$ test.

Oriental lineage and was subsequently lost at least twice independently. Moreover, it appears that the relative position and shape of the hg1 muscle within the thoracic box are largely conserved across the phylogeny (Fig. S4).

## Sexual size dimorphisms in the *Drosophila* wing musculature are not limited to hg1

The evolutionary transitions in hg1 described above may have been associated with changes in male courtship song (e.g., gain or loss of sine song) or other sexually dimorphic wing-based behaviors. Alternatively, evolutionary transformations in the sexual size of hg1 may have been accompanied by compensatory changes in other wing muscles that would abrogate evolutionary changes in behavior (e.g., if the behavior was under stabilizing selection). There is precedent for this hypothesis, as hg1 is one of four control wing muscles (hg1–4) that insert into a common tendon that influences the mechanical properties of the posterior notal wing process at the wing hinge (Fig. 1); moreover, all four hg muscles are active during *D. melanogaster* courtship song (*O'Sullivan et al., 2018*). Given the potential functional overlap among the hg muscles, loss of the size dimorphism in hg1 may have been associated with a compensatory gain of a size dimorphism in another hg wing muscle,

or vice versa. To test this hypothesis, we analyzed the evolutionary history of sexual size dimorphism in the hg2, hg3, and hg4 muscles.

Analyses of hg2, hg3, and hg4 muscle size from males and females of the 19 species studied above revealed that the hg2 and hg3 muscles evolved a size dimorphism multiple independent times in the *Drosophila* phylogeny (Figs. 4A, 4B, Fig. S1 Table S4). Hg4, however, has remained anatomically monomorphic through *Drosophila*'s evolutionary history (Fig. 4C). The magnitude of the size dimorphisms in hg2 and hg3 are generally lower than those observed in hg1 (Fig. 2). Notably, if the gain or loss of a size dimorphism in hg1 (Fig. 3) was compensatory to an anatomical transition in another hg muscle, we would expect to find either hg2, hg3, or hg4 to be male-enlarged where hg1 is monomorphic or vice versa. However, none of the hg muscles display such a pattern, indicating that evolutionary changes in hg1 were most likely not compensatory to changes in other hg wing muscles.

Our analyses reveal a surprising amount of species variation in sexual size dimorphism in the *Drosophila* wing musculature. Ancestral state reconstruction of hg2's sexual size dimorphism (Fig. 4A) indicates that the ancestor of the Oriental lineage most likely had a monomorphically sized hg2 muscle (probability of a monomorphic ancestor = 0.999). The hg2 muscle appears to have subsequently evolved a size dimorphism in members of the *suzukii* species subgroup (i.e., *D. mimetica*, *D. prostipennis*, and *D. biarmipes*). Outside of the Oriental lineage, hg2 evolved a size dimorphism at least once in the *ananassae* species group. It is noteworthy that unlike the dimorphisms in hg1, which were always male-enlarged, hg2 has evolved male- and female-enlarged size dimorphisms.

Ancestral state reconstruction of hg3 and hg4's sexual size dimorphism (Fig. 4B, Fig. 4C) reveals that a monomorphic hg3 and hg4 muscle was most likely present throughout the basal lineages and the Oriental lineage. A size dimorphism in hg3 evolved independently four times in *D. elegans*, *D. yakuba* and *D. prostipennis* (male-enlarged), and in *D. erecta* (female-enlarged) (Fig. 4B). None of the taxa sampled display a dimorphic hg4 muscle (Fig. 4C). Like hg1, the relative position and shape of the hg2–4 muscles within the thorax have not changed in any obvious way (Fig. S4).

## DISCUSSION

Here, focusing on a set of four control wing muscles in *Drosophila*, we report that the *Drosophila* wing musculature has undergone a notable number of evolutionary transitions in sexual size dimorphism (data summarized in Fig. 5). In ~25 Mya since the *melanogaster* species group arose, we uncovered a total of 11 independent transitions among the four wing muscles we studied: 8 gains of dimorphic size, both male- and female-enlargement; 2 reversions to monomorphism; and 1 conversion from male- to female-enlargement.

Our analyses also revealed that different wing muscles vary in their proclivity to evolve a size dimorphism. Of the four wing muscles we analyzed, three (hg1–3) have gained and lost size dimorphisms several times, whereas one (hg4) has remained monomorphic throughout the phylogeny. This pattern of evolutionary change could be shaped by functional constraints acting upon some wing muscles that select against the evolution of a

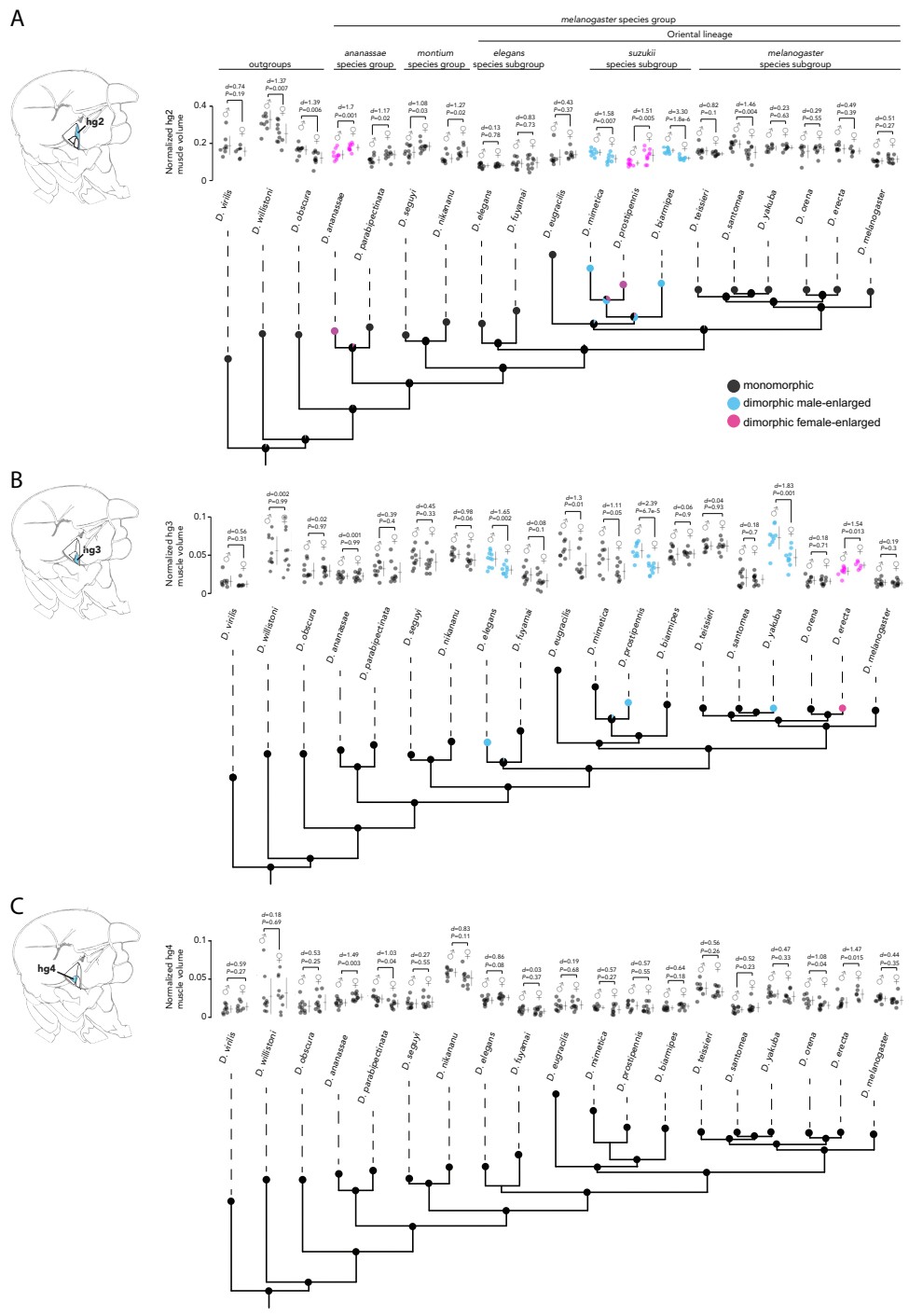

**Figure 4  Evolutionary history of sexual size dimorphism in the hg2 (A), hg3 (B), and hg4 (C) wing muscles of *Drosophila*.** In each panel, a maximum likelihood tree showing phylogenetic relationships among 19 *Drosophila* species in shown below. Circles at internal nodes (continued on next page...)
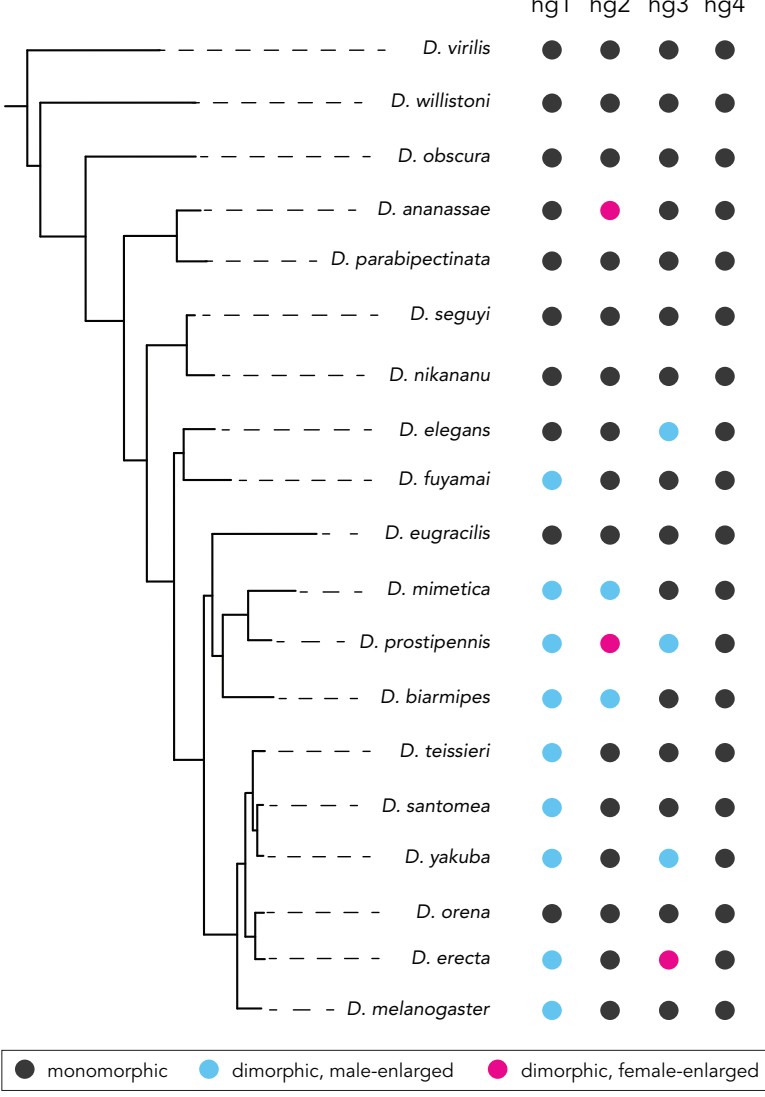

**Figure 5 Summary of species with or without a sexual size dimorphism in the hg1–4 muscles.** Species with a monomorphically sized muscle are shown in black; and species with a dimorphically sized muscle are shown in blue (male-enlarged) and red (female-enlarged).

size dimorphism. The nature of these constraints is unclear, but recent work on the control of Dipteran flight may offer some insight. In *Drosophila*, wing control muscles fall into two anatomically and functionally distinct groups (*Lindsay, Sustar & Dickinson, 2017*): one group of relatively large 'phasic' muscles are active during big, transient changes in wing motion like those seen during rapid flight maneuvers; a second group of relatively small 'tonic' muscles influence wing motion in a continuous, fine-scaled, and graded fashion, enabling flies to fly stably and remain on course in the face of aerodynamic perturbations. It is noteworthy that hg1 and hg3, both of which evolved size dimorphisms, are 'phasic' muscles, whereas hg4, which has remained monomorphic in the phylogeny, is a 'tonic' muscle. This suggests the possibility that 'tonic' muscles are generally recalcitrant to gaining a sexual size dimorphism perhaps due to their functional role during flight. 'Phasic' muscles may lack these constraints.

Most control wing muscles are associated with one of several interconnected sclerites at the wing hinge, and contribute to wing kinematics by regulating the mechanical configuration of these sclerites (*Deora, Gundiah & Sane, 2017*). Evolutionary changes in wing muscle anatomy may thus have been accompanied by morphological transitions in the exoskeletal elements and biomechanics of the wing hinge—a hypothesis that would be interesting to investigate in future studies.

Evolutionary transitions in dimorphic wing muscle size may have contributed to species variation in *Drosophila* courtship song, or other sexually dimorphic wing-based behaviors (*e.g.,* aggression). Given the role of hg1 and its sexual identity in the regulation of *D. melanogaster* sine song, the evolutionary changes in hg1's dimorphic size may have been related to divergence of sine song. Consistent with this hypothesis, species such as *D. virilis*, *D. willistoni*, and *D. ananassae,* all of which have a monomorphically sized hg1 muscle (Fig. 3), lack sine song during male courtship (*Tomaru & Yamada, 2011*); on the other hand, *D. biarmipes*, *D. teissieri*, and *D. erecta* males all sing sine song (*Tomaru & Yamada, 2011*; *Mazzoni, Anfora & Virant-Doberlet, 2013*) and have a male-enlarged hg1 muscle (Fig. 3). Furthermore, although the sister species *D. yakuba* and *D. santomea* have a male-enlarged hg1 muscle (Fig. 3) despite having lost sine song (*Watson, Rodewald & Coyne, 2007*), the magnitude of the size dimorphism is notably weak relative to most other species with a dimorphic hg1 muscle (Fig. 3). These findings suggest that the acquisition of a dimorphic hg1 muscle in the Oriental lineage may have been associated with the emergence of sine song in *Drosophila*.

To evolve a dimorphically sized wing muscle, the developmental pathways that regulate muscle size must have been integrated with the sexual differentiation pathway in flies. Our previous work demonstrated that hg1's dimorphic size development in *D. melanogaster* is regulated by sex-specific transcription factors encoded by the sexual differentiation gene, *doublesex* (*Shirangi, Stern & Truman, 2013*). This suggests that dimorphic wing muscle size evolved in part by a gain of *doublesex* expression in the developing wing muscle. It will be interesting to test this hypothesis in future work by probing *doublesex* expression in the developing wing musculature across the *Drosophila* phylogeny.

## CONCLUSIONS

The wing muscles of *Drosophila* have repeatedly evolved sexual size dimorphisms suggesting that evolutionary changes in wing muscle anatomy may have contributed to species variation in sexually dimorphic wing-based behaviors like courtship song. Not all wing muscles have evolved a sexual size dimorphism, suggesting that wing muscles vary in the selective pressures that drive a sexual size dimorphism.

## ACKNOWLEDGEMENTS

We thank Todd Jackman and Aaron Bauer for their tutelage on the phylogenetic analyses; and Yun Ding, Ella Preger-Ben Noon, Shannon Ballard, and Todd Jackman for comments on the manuscript.

### Funding

This work was initially supported by Dr. James Truman (HHMI) and later by start-up funds from Villanova University. The funders had no role in study design, data collection and analysis, decision to publish, or preparation of the manuscript.

### Grant Disclosures

The following grant information was disclosed by the authors:
Villanova University.

### Competing Interests

The authors declare there are no competing interests.

### Author Contributions

- Claire B. Tracy and Troy R. Shirangi analyzed the data, conceived and designed the experiments, performed the experiments, prepared figures and/or tables, authored or reviewed drafts of the paper, and approved the final draft.
- Janet Nguyen and Rayna Abraham performed the experiments, prepared figures and/or tables, carried out experimental methods, and approved the final draft.

### Data Availability

The raw normalized muscle volumes for hg1-4 muscles from the 19 species under study are available in File S2.

### Supplemental Information

Supplemental information for this article can be found online at http://dx.doi.org/10.7717/peerj.8360#supplemental-information.

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
