# Peer review of "Evolution of sexual size dimorphism in the wing musculature of Drosophila"

_PeerJ, doi:10.7717/peerj.8360_

## Round 0.1 · original submission · Minor Revisions

I think the small edits in text could be taken on without too much bother. The request from Reviewer One for some raw data images would help strengthen what is a really elegant study.

·

Basic reporting

This is a well written manuscript that is easily read and understood and conforms to PeerJ standards. The aims and conclusions of the study are clearly stated and well referenced. The figures are excellent quality and readily interpreted with fully explanatory legends. The raw numerical data has been supplied but not any of the original morphology studies. As such the numerical data do corresponds with the data used in the manuscript but the lack of the original morphological data is a bit of an issue.

Experimental design

The manuscript presents an interesting and original piece of primary research that fits comfortably with the scope of the journal. Whilst the work is primarily descriptive it addresses a fascinating and important area of evolutionary biology. The core approach is to use comparative and quantitative morphology to shed light on a subject that will be of interest not only to a specialist audience but also a more general scientific and even lay audience. The quality of the work is exceptional, it is a simple concept executed well and rigorously analysed. The methodologies are adequately described with sufficient detail. I am not qualified to comment on the phylogenetic methodologies used but I am fully familiar with morphometric studies. If I were to have a criticism is that the paper does not show any images of the muscles studied and it isn’t possible to assess what the muscles look like nor see how they differ in size and shape. I am sure that the shape of the muscles may also provide interesting insights into the evolution of song. The paper could be improved by the inclusion of a figure showing how the muscles differ in proportion and shape even in there is nothing interesting in their shapes.

Validity of the findings

The lack of morphology aside the results are still very interesting and provide insight into a much-studied aspect of Drosophila biology. it is without doubt an interesting and important contribution that will stimulate further study.

I have no issues with grammar or the written English and have no changes to recommend.

In summary the work is simple and elegant in concept, excellent in execution and presentation. I would recommend acceptance with a suggestion of the additional anatomical figure.

Additional comments

I would encourage the authors to consider showing some of the morphological data, even if the analysis is qualitative.

Reviewer 2 ·

Basic reporting

Well written article

Experimental design

Well written.

Validity of the findings

Strong results, but discussions speculative in parts.

Additional comments

In this manuscript, Tracy et al provide fascinating comparative insights into the structures of 4 steering muscles (hg1-4 muscles) which are associated with the actuation of the posterior notal wing process in Drosophila species. Previous work by Shirangi et al discovered the role of one of these muscles (hg1) in generating the ‘sine’ song during courtship in the case of D. melanogaster. This muscle was found to be sexually dimorphic, as it is much larger in males than in females.

In this manuscript, the authors extend this study by conducting a thorough exploration of size variation in hg1-4 muscles across 19 Drosophilid species. Their study finds a great degree of species-specific variation and size dimorphism in these muscles. Of these, hg1 muscles are most size dimorphic in the Oriental lineage (which diversified most recently). In comparison, the hg2 muscles are usually monomorphic, but in a few cases show size dimorphism and female enlargement. Similarly, hg3 muscles also show size dimorphism in some clades, but are usually monomorphic. In contrast to these, the hg4 muscle is entirely monomorphic in all species.

The authors discuss what these findings might mean in terms of their role in courtship song diversification and speciation. It must be said that the discussion is very speculative and often veers into the ‘why’ territory, but I did not feel that any of it was unreasonable. Indeed, it is rare to see studies that throw light on evolution of behavioural traits (in this case, courtship song) using carefully conducted morphological studies. Because the authors had already developed insights about hg1’s role in courtship song, their study and inferences are convincing, the methods are sound, and the paper is very well-written. As such I have no major comments or criticisms of the paper, and recommend it for publication.

I did note a few minor points that occurred to me while reading the manuscript:

1. Line 256-260: “It is also possible that …. may have evolved to prevent dimorphism in a monomorphic behaviour”. This entire paragraph is much too speculative and should be omitted in my opinion. It is true that steering muscles also play a role in flight, and hence any ‘song-specific’ changes in their structure must potentially trade-off with flight performance. The only problem is: we don’t fully understand their role in flight. A recent paper by Lindsay et al (which is cited in this paper) demonstrated the functional organization of these steering muscles into two specific functional groups - the larger phasically activated muscles that actuate the wing movements required during sudden turns, and the smaller tonically-activated muscles whose role is subtler but continuous. Lindsay et al find that hg1 and hg3 are phasic muscles, whereas this paper shows significant size dimorphism in these muscles, which may in principle influence flight. The zeroth-order experiment would be to knock out these muscles individually or alter their relative sizes, and test the effect on flight performance which I hope will be conducted in due course. Also, the authors often write sentences of the form “X evolved for Y purpose”. Such sentence construction is problematic and should be avoided, as it suggests directionality and purpose to evolution.

2. The entire discussion on song production emphasizes the role of muscles, but entirely ignores exoskeletal structures and their biomechanics which also must co-evolve with these muscles. Actuation of these inter-connected skeletal structures dictates how wings move in response to muscle activity. These have not been dealt with in the paper, but a sentence or two to convey their importance may be worth including.

3. Line 254: “raise a hypothesis”? Sounds awkward.

4. Fig 2: I felt that choosing a Cohen's d value of 1.5 as the cut-off between dimorphic and monomorphic was somewhat arbitrary, although I do not entirely disagree with that choice in cases for which there was a clear gap in size variation. However, in the case of hg1, there is no clear gap and hence the difference between mono- and dimorphic seems trickier to nail. Can the authors please comment on this?

---

## Round 0.2 · Minor Revisions

The work is great and the data provided, with the rebuttal, of the highest quality. To ensure this lovely work is fully supported and recognised for its excellence please include one or two datasets showing raw confocal data.

Many thanks,

Darren

---

## Round 0.3 · accepted · Accept

I really like the changes and the additional data figures to show both original phalloidin scans and segmentations. Excited by the future work it will open up.